# High Throughput Genetic Characterisation of Caucasian Patients Affected by Multi-Drug Resistant Rheumatoid or Psoriatic Arthritis

**DOI:** 10.3390/jpm12101618

**Published:** 2022-09-30

**Authors:** Paola Tesolin, Francesca Eleonora Bertinetto, Arianna Sonaglia, Stefania Cappellani, Maria Pina Concas, Anna Morgan, Norma Maria Ferrero, Alen Zabotti, Paolo Gasparini, Antonio Amoroso, Luca Quartuccio, Giorgia Girotto

**Affiliations:** 1Department of Medicine, Surgery and Health Sciences, University of Trieste, 34149 Trieste, Italy; 2Department of Medical Sciences, University of Turin, and Immunogenetic and Transplant Biology Service, University Hospital “Città della Salute e della Scienza”, 10124 Turin, Italy; 3Division of Rheumatology, Department of Medicine (DAME), ASUFC, University of Udine, 33100 Udine, Italy; 4Institute for Maternal and Child Health—IRCCS, Burlo Garofolo, 34137 Trieste, Italy

**Keywords:** rheumatoid arthritis, psoriatic arthritis, multi-drug resistance, whole-exome sequencing, SNPs-array, HLA typing

## Abstract

Rheumatoid and psoriatic arthritis (RA and PsA) are inflammatory rheumatic disorders characterised by a multifactorial etiology. To date, the genetic contributions to the disease onset, severity and drug response are not clearly defined, and despite the development of novel targeted therapies, ~10% of patients still display poor treatment responses. We characterised a selected cohort of eleven non-responder patients aiming to define the genetic contribution to drug resistance. An accurate clinical examination of the patients was coupled with several high-throughput genetic testing, including HLA typing, SNPs-array and Whole Exome Sequencing (WES). The analyses revealed that all the subjects carry very rare HLA phenotypes which contain HLA alleles associated with RA development (e.g., HLA-DRB1*04, DRB1*10:01 and DRB1*01). Additionally, six patients also carry PsA risk alleles (e.g., HLA-B*27:02 and B*38:01). WES analysis and SNPs-array revealed 23 damaging variants with 18 novel “drug-resistance” RA/PsA candidate genes. Eight patients carry likely pathogenic variants within common genes (*CYP21A2, DVL1*, *PRKDC*, *ORAI1*, *UGT2B17*, *MSR1*). Furthermore, “private” damaging variants were identified within 12 additional genes (*WNT10A*, *ABCB7*, *SERPING1*, *GNRHR*, *NCAPD3*, *CLCF1*, *HACE1*, *NCAPD2*, *ESR1*, *SAMHD1*, *CYP27A1*, *CCDC88C*). This multistep approach highlighted novel RA/PsA candidate genes and genotype-phenotype correlations potentially useful for clinicians in selecting the best therapeutic strategy.

## 1. Introduction

Rheumatoid arthritis (RA) and psoriatic arthritis (PsA) are inflammatory rheumatic disorders affecting ~1–2% of the European population [1,2] and highly impacting the quality of life. 

RA is more common among women and has an onset ranging from the fifth to the sixth decades of life [1,3]. Clinically, RA patients manifest symmetrical peripheral polyarthritis and inflammatory pain, but disease progression may lead to damaged joint deformity and increased risk of cardiovascular diseases. Further, RA is an autoimmune disease; indeed, patients often produce different autoantibodies, such as the rheumatoid factor (RF) and the anticitrullinated peptide/protein antibodies (ACPA), which are often used as RA classification criteria [1,3]. 

As regards PsA, ~40% of the subjects with psoriasis eventually develop it, with the same frequency in both genders and varying ages of onset. PsA belongs to the spectrum of spondyloarthropathy, and it is generally classified as an RF negative inflammatory arthritis in the presence of psoriasis [2,4].

Overall, a multifactorial etiology characterizes both the diseases, with environmental and genetic factors influencing their onset, progression, clinical variability and, eventually, the drug response [3,4,5]. 

Regarding the genetic component of RA and PsA, HLA genes seem to play an essential role, and the presence of specific HLA alleles is defined as a predisposing factor for both diseases [6,7]. In particular, HLA-DRB1 alleles have been strongly associated with the risk of developing RA. Generally, the RA-associated HLA-DRB1 alleles are characterised by a sequence of five amino acids, called the “shared epitope”, which might lead to the incorrect presentation of autoantigens to T cells by the antigen-presenting cells (APCs), causing an autoimmune response. Further, patients carrying the shared epitope alleles are predisposed to more destructive joint disease and increased mortality [6]. In particular, HLA-DRB1*04 is the allele that has been more frequently associated with RA development. However, other alleles, such as HLA-DRB1*01 and HLA-DRB1*10:01 have a role in RA [1]. 

Regarding PsA-predisposing alleles, it has been proven that HLA class I genes have a critical role. Interestingly, each allele is associated with different clinical features. In particular, HLA-B*39:01, HLA-B*38:01, and HLA-B*08:01 are risk factors for the development of PsA [7], but HLA-B*27 is associated with more severe forms as well as with specific symptoms, such as spondylitis and uveitis [8,9].

So far, few non-HLA genes have also been associated with the development of both diseases. For example, it has been established that HLA-DRB1 alleles and pathogenic variants within the *PTPN22* gene are responsible for 50% of the genetic component of RA in European patients [1]. Regarding PsA, *IL23R*, *TNFAIP3*, *IL12B*, and *PTPN22* gene variants are defined as risk factors [4].

Patients affected by RA/PsA can follow different therapeutic regimes with synthetic and biological Disease-Modifying Antirheumatic Drugs (DMARDs) [10,11]. Briefly, RA patients are initially treated with conventional synthetic (cs) DMARDs (i.e., methotrexate (MTX), leflunomide, sulfasalazine); in the case of therapeutic failure, targeted synthetic (ts) DMARDs (i.e., JAK inhibitors) and biological (b) DMARDs can also be employed. To date, several bDMARDs, and their biosimilar (bsDMARDs), targeting different pathways, are available (i.e., anti-TNF, anti-IL6 receptor, anti-IL-1ra abatacept, rituximab). Glucocorticoids (GC) are often co-administered with various therapies [10]. On the other hand, PsA-specific therapeutic regimens often include anti-IL17 and anti-IL12/IL23 therapies. Anti-TNF drugs are also commonly administrated

Despite the continuous development of new and innovative therapeutic approaches, about 5–10% of patients still display poor responses to the therapies [12]. According to the European Alliance of Associations for Rheumatology (EULAR) recommendations, the lack of efficacy can be classified as primary failure, assessed 6 months after the start of the therapy, or secondary failure, which develops in initial responders [6,10]. Patients are commonly defined as “difficult to treat” after the failure of csDMARDs and of at least two bDMARDs with different mechanisms of action [13]. Nevertheless, the definition of “drug-resistance” in inflammatory rheumatic diseases is often still challenging, but it is a crucial step for precision medicine. 

Identifying the molecular basis of drug resistance could be extremely relevant, guiding the selection of the proper therapeutic strategies and avoiding unnecessary and potentially harmful treatments. 

The introduction of next-generation sequencing (NGS) technologies has already improved the study and management of several genetic disorders, and it is expected to be useful also in unveiling the molecular basis of many other traits, including drug resistance (15). In particular, NGS will allow us to shed light on unknown disease mechanisms and lead to the selection of the appropriate drug for each patient [14,15].

Taking advantage of the utmost genomic approaches, here we describe a cohort of eleven individuals affected by RA or PsA, characterised by multidrug resistance due to persistent inflammation, carefully selected from a cohort of 649 RA/PsA patients on treatment with biologic agents or small molecules [16]. We performed HLA typing, SNPs-array (to detect large structural variants) and Whole Exome Sequencing (WES) high-throughput genetic analyses in order to better clarify the genetic contribution of each subject to drug resistance, aiming to provide clinicians with therapeutic suggestions and eventually identify novel molecular targets for drug development. 

## 2. Materials and Methods

### 2.1. Patients Enrolment

Eleven Caucasian patients affected by RA (N = 9) and PsA (N = 2) and classified as “difficult-to-treat” were enrolled at the Division of Rheumatology, Department of Medicine, of the University of Udine (Udine, Italy) from February to April 2021. The inefficacy of the employed treatment was defined whenever the patient did not reach the target (i.e., remission or reduced disease activity) by six months, and was divided into primary inefficacy (lack of response) or secondary (loss of response), following the EULAR recommendation [10,13]. All the individuals provided written informed consent for the analyses, and the research was conducted according to the ethical standard defined by the Helsinki Declaration.

### 2.2. DNA Extraction and Quantification 

Genomic DNA (gDNA) was extracted from peripheral whole blood samples using the QIAsymphony^®^SP instrument with QIAsymphony^®^DNA Midi kit (Qiagen, Venlo, The Netherlands), and DNA concentration was measured using Nanodrop ND 1000 spectrophotometer (NanoDrop Technologies Inc., Wilmington, DE, USA). 

### 2.3. HLA Typing

We typed 11 HLA loci (i.e., HLA-A, HLA-B, HLA-C, HLA-DRB1, HLA-DRB3,4,5, HLA-DQB1, HLA-DQA1, HLA-DPB1, HLA-DPA), with NGS technologies. The HLA typing was performed using commercially available reagents (GenDX, Utrecht, The Netherlands) and a MiSeq platform (Illumina, San Diego, CA, USA). Data were analysed using the NGSengine Software (version 2.25.0; GenDx, Utrecht, The Netherland) and the IPD-IMGT/HLA Database (Release 3.49) [17].

#### Evaluation of the Frequency of Patient’s HLA Phenotypes and Haplotypes

We performed preliminary research in the worldwide registry of Hematopoietic Stem Cells (HSC) and core blood donors to determine the frequencies of the patients’ HLA phenotypes in the world population. At the time of the research (May 2022), the collection counted 39,942,217 entries regarding the HLA typing of each donor. Thus, a simulation was performed searching for potential matches between the HLA phenotypes of the patients and all the donors. Briefly, an authorised operator filled in the form with the HLA-A, -B, -C, -DRB1, -DQB1, -DPB1 typing of each patient by employing the IBMDR (Italian Bone Marrow Donor Registry, version XXIV, August 3rd 2021) software. The software suggested donors with different compatibility levels. Finally, only donors with 10/10 compatibility at five loci were selected. Furthermore, A Chi-square test by 2 × 2 contingency table was used to compare the frequency of HLA haplotypes between patients and an Italian Control population [18].

### 2.4. SNPs-Array

SNPs-array analysis was performed using the Infinium Global Screening Array-24 v3.0 BeadChip (Illumina Inc., San Diego, CA, USA), containing 654,027 markers. A total of 200 ng of gDNA for each sample was processed according to the manufacturer’s instructions. Normalisation of raw image intensity data, genotype clustering, and individual sample genotype calls were performed using Illumina’s Genome Studio software v2.0.3 (cnvpartition 3.2.0). The Copy Number Variants (CNVs) were mapped to the human reference genome hg19 and annotated with UCSC RefGene. Allele detection and genotype calling were performed with Genome Studio software by evaluating B allele frequencies (BAFs) and log R ratios.

The frequency in the general population was defined by employing the Database of Genomic Variants (http://dgv.tcag.ca/dgv/app/home, accessed on 5 August 2022), aiming to exclude those that could be considered polymorphic. Further, the database DECIPHER (https://www.deciphergenomics.org/, accessed on 5 August 2022) was used to define whether the CNV was already associated with other conditions. 

### 2.5. WES

WES was carried out on an Illumina NextSeq 550 instrument (Illumina Inc., San Diego, CA, USA), following the manufacturer’s instructions. First, the enzymatic fragmentation of 50 ng of gDNA was performed, followed by end repair and dA-tailing reactions. Subsequently, each fragment was ligated to a universal adapter and amplified using the Unique Dual Index primers (Twist Bioscience, South San Francisco, CA, USA). Finally, genomic libraries were generated using the Twist Human Core Exome + Human RefSeq Panel kit (Twist Bioscience, South San Francisco, CA, USA). In conclusion, the hybridised fragments were captured, amplified and sequenced. The process leads to the generation of FASTQ files, which are processed through a custom pipeline (Germline-Pipeline), developed by enGenome srl. This workflow leads to the creation of a VCF file containing germline variants, such as Single Nucleotide Variants (SNVs) and short insertion/deletions (INDELs). VCF files were analysed on enGenome Expert Variant Interpreter (eVai) software (evai.engenome.com) [19]. As a final step, variants were selected by applying several filters, such as quality score (QUAL) > 20 and Minor Allele Frequency (MAF) < 0.001. Further, the pathogenicity of known genetic variants was evaluated using ClinVar (http://www.ncbi.nlm.nih.gov/clinvar/ (accessed on 5 August 2022) and The Human Gene Mutation Database (http://www.hgmd.cf.ac.uk/ac/index.php, accessed on 5 August 2022). Several in-silico tools, such as PolyPhen-2 [20], SIFT [21], Pseudo Amino Acid Protein Intolerance Variant Predictor (for coding variants SNVs/INDELs) (PaPI score) [22] and Deep Neural Network Variant Predictor (for coding/non-coding variants, SNVs) (DANN score) [23] were used to evaluate the pathogenicity of novel variants. As the last step, on a patient-by-patient basis, the correlation between the variants and the phenotypes was discussed, and the related literature was evaluated. The most compiling variants were confirmed by direct Sanger sequencing (primers and thermic protocols are available upon request).

#### Statistical Analysis

The variants identified in the RA/PsA group were checked in a Whole-Genome Sequencing (WGS) control cohort composed of 377 healthy individuals matched by sex and age with the patients. In detail, phenotypic and genetic data of the control subjects were already available at the Institute for Maternal and Child Health “Burlo Garofolo” [24]. The subjects from the WGS cohort were selected as controls if their anamnestic form did not report any clinical feature that could be linked to “difficult to treat” RA or PsA. Notably, most RA and PsA drug responder patients usually reach the therapeutic goals with the first or second line of treatment employed; thus, no additional drug would have been administered (i.e., only the response to a few drugs is known). Since there are no validated biomarkers of response to the available DMARDs and the response to most available drugs would have been unknown, a control cohort including RA and PsA patients was not included.

## 3. Results

### 3.1. Patients’ Clinical Characteristics 

The clinical features of the patients enrolled in the present study and the complete list of administered therapies are summarised in Table 1. The selected patients display a rare and characteristic phenotype (i.e., extensive resistance to several ts-/b-DMARDs), which was identified in 1.7% of the subjects (11/649) of the initially considered RA and PsA population. 

### 3.2. HLA Typing

The complete results of the HLA typing are available in Appendix A. Table 2 displays the HLA-B, HLA-C and HLA-DRB1 alleles detected in the patients, which have been previously associated with the development of the two diseases. 

All the patients carry at least one HLA-DRB1 allele associated with the risk of developing RA, such as DRB1*10:01, DRB1*01:01 and DRB1*01:02, DRB1*04:01P and DRB1*04:08:01 [1]. Two patients of the cohort (SP5 and SN4) have the HLA-DRB1*13:03 and HLA-DRB1*03:01 alleles, which have been described as a RA risk factor [6,25] even if not containing the shared epitope. 

Regarding PsA predisposing HLA-alleles, SP1, SN1, SN4, and SN6 carry one risk allele, such as HLA-B*38:01, B*08:01, B*39:01 and B*27:02:01:01 [7], while SP5 has two PsA predisposing alleles (i.e., HLA-B*08:01P and HLA-B*38:01P) [7]. Furthermore, patient SN3, who is affected by PsA, carries the allele HLA-C*06:02:01:01, which has been linked with the development of PsA/psoriasis [7]. 

Thus, six out of eleven patients (55%) display a double genetic predisposition to the development of both RA and PsA/psoriatic disease.

Furthermore, our cohort is enriched in a particular rare ancestral haplotype (i.e., B*14:02; C*08:02; DRB1*01:02; DQB1*05:01. When comparing the frequency of this ancestral haplotype between the patients and the control Italian population [18], a high significant difference was found (*p*-value 0.00001). In detail, three subjects (SP4, SN2 and SN3) carry this HLA haplotype, which has been associated with the development of autosomal dominant frontal fibrosing alopecia, a disorder characterised by the recession of the hairline, co-occurrence of autoimmune diseases (e.g., thyroid disorders and Sjögren’s syndrome) and sex-hormonal imbalance [26]. This haplotype is known for being in linkage disequilibrium with the c.844G>T, p.Val282Leu variant within the *CYP21A2* (NM_000500.7) gene [27], whose presence at the heterozygous state has been confirmed in the three patients. The *CYP21A2* gene is located within the human leukocyte antigen complex, and it encodes for the enzyme steroid 21-hydroxylase that converts pregnenolone into cortisol [26,27]. 

In addition, to assess the frequency of the HLA phenotypes of the patients analysed, we simulated a match search between all the potential donors from the HSC and cord blood registries. The results of this analysis are reported in Table 3. Interestingly, the HLA haplotypes of the eleven patients are characterised by a particularly low frequency (<0.027%) in the world population.

### 3.3. SNPs-Array

SNPs-array carried out in the eleven patients revealed a particularly interesting CNV in the genome of patient SN3. In detail, she carries a heterozygous deletion of 71077 base pairs (Chr8:15950391-16021468), with a frequency of 0,4% in the general population [28], which affects the gene *MSR1*, from exon 6 to 9. The gene encodes for a class A macrophage scavenger receptor, which plays a role in osteogenic differentiation and autoimmunity [29,30]. 

### 3.4. WES

WES data analysis was carried out using a hypothesis-free approach and revealed that five patients (SP1, SP2, SP3, SP4, SN1, SN4) carry rare (MAF < 0.001) and highly impacting variants in common genes (*DVL1*, *PRKDC*, *UGT2B17*, *ORAI1*). Furthermore, eight patients (SP1, SP2, SP4, SN2, SN3, SN4, SN5, SN6) carry “private” and, likely pathogenic, variants within 13 genes (*WNT10A*, *ABCB7*, *MSR1*, *SERPING1*, *GNRHR*, *NCAPD3*, *CLCF1*, *HACE1*, *NCAPD2*, *ESR1*, *SAMHD1*, *CYP27A1*, *CCDC88C*) (see Table 4). 

Regarding the common genes mentioned above, the following results were identified: (a) two unknown missense variants (c.780C > G; p.His260Gln and c.1148C > A; p.Ser383Tyr) within the *DVL1* gene which is an essential member of the Wnt signalling pathway [31], in patients SP1 and SP4; (b) two novel missense variants (c.7790G > A; p.Arg2597Gln and c.5983G > A; p.Val1995Ile) in *PRKDC* gene, which encodes for the catalytic subunit of kinase protein that interacts with the transcription factor autoimmune regulator in T-cell [32] in patients SP4 and SP3; (c) two novel damaging variants (c.144_150dupCGCCGTC; p.Pro51fs and c.698C > T; p.Pro233Leu) within *ORAI1*, a primary regulator of calcium levels in the T-cells [33,34,35], in patients SP4 and SN1; (d) two unknown damaging variants (c.1237G > T; p.Ala413Ser and c.1158C > A; p.Tyr386*) within *UGT2B17* gene, which is essential for the metabolism of sex hormones and the clearance of ~25% of common medications [36], in patients SP3 and SN4.

Moreover, two subjects (SN3 and SN5) carry variants in two genes belonging to the same family. In detail, patient SN3 has the variant c.1712A>C; p.His571Pro within the *NCAPD3* gene, while subject SN5 carries the variant c.860C>G; p.Pro287Arg within *NCAPD2*. Both genes have a role in releasing pro-inflammatory cytokine and modulating the NF-κB signalling pathway [37]. 

Furthermore, all the patients, except SP3, SP5 and SN1, carry likely damaging variants in private genes that, according to literature data, may have a role in RA/PsA and drug response (Table 4). 

Briefly, SP1 carries a known pathogenic variant (c.321C>A; p.Cys107*) within *WNT10A*, which is a gene involved in the Wnt signalling and associated with B-cell dysfunctions [38,39].

In patient SP2, two novel variants were identified in the genes *ABCB7* (c.1767T>G; p.His589Gln) and *MSR1* (c.260C>G; p.Ser87*). *ABCB7* encodes for a mitochondrial iron transporter which regulates the levels of reactive oxygen species and inhibits NF-kB signalling [40]. Further, this patient carries a nonsense variant within *MSR1*, the same gene previously reported as deleted in subject SN3.

Patient SP4 carries a missense variant (c.1198C>T; p.Arg400Cys) within the *SERPING1* gene, which encodes for a serine protease inhibitor of the complement system (C1 inhibitor) [41,42]. The variant has already been classified as damaging as it causes protein misfolding under mild stress conditions [41,42].

In subject SN2, we detected a known pathogenic missense variant (c.785G>A; p.Arg262Gln) within the *GNRHR* gene, which encodes for a gonadotropin-releasing hormone receptor [43,44].

Patient SN3 carries a heterozygous missense variant (c.226C>G; p.Pro76Ala) within *CLCF1*, a member of the IL-6 family of cytokines with a role in bone formation [45].

Subject SN4 carries an unknown missense variant (c.217G>A; p.Ala73Thr) within *HACE1*, an inhibitor of the TNF-stimulated NF-kB pathway [46]. 

In the only male subject of the cohort, patient SN5, a novel missense variant (c.715G>T; p.Ala239Ser) was identified in *ESR1*, which encodes for an estrogen receptor and ligand-activated transcription factor [47]. 

Finally, in subject SN6, three private missense variants were detected within the *SAMHD1* (c.997C>T; p.Arg333Cys), *CYP27A1* (c.1435C>T; p.Arg479Cys) and *CCDC88C* (c.5102G>T; p.Arg1701Leu) genes. *SAMHD1* has a role in inflammation and autoimmunity [48,49,50], while *CYP27A1* is involved in the synthesis of oxysterol 27-hydroxycholesterol (27HC), a selective modulator of estrogen receptors with immunomodulatory roles [51]. Finally, *CCDC88C* regulates the Wnt signalling pathway and the inflammatory responses [52]. 

None of these 21 variants has been identified in the WGS control cohort of 377 individuals matched for age and sex [24].

## 4. Discussion

RA and PsA are relatively common inflammatory rheumatic disorders characterised by a multifactorial etiology, thus, genetic and environmental factors play a role in their onset, progression, severity and drug responses [1,4,5,6,8]. The diseases arise from the simultaneous dysregulation of multiple pathways, and several targeted therapies with different mechanisms of action have been introduced into clinical practice [10,11]. Despite the continuous development of novel therapeutic strategies, drug response is highly variable [14] and, in a few cases, such as those here described, patients may not experience any benefit from the numerous administered therapies. This could be ascribed to (1) a lack of pharmacogenomic guidelines implementations or (2) the administration of drugs not directed against the optimal molecular targets. In the case of the highly selected cohort here described, the most probable cause is the second one, further supported by the fact that the patients displayed a type I inefficacy for most of the administered drugs (51% of the b-DMARDs and ts-DMARDs). Furthermore, the patients were characterised by a particularly rare phenotype (i.e., “difficult-to-treat” due to extensive resistance to many ts-/b-DMARDs), suggesting that the drug resistance they experienced may have a genetic base. Importantly, “a true refractory phenotype” was analysed, which accounts for less than 3% of the patients of the cohort, in particular for RA [53]. Thus, several high-throughput genetic screenings (such as HLA typing, SNP-array and WES) have been applied to detect genetic variants responsible for the observed phenotype. 

The first relevant findings are related to HLA typing. The patients’ HLA phenotypes highlighted from the analyses are notably rare (<0.027%) in the world population, suggesting their possible association with the peculiar phenotype displayed by the patients. Furthermore, all the patients are predisposed to RA development, and 55% of them also carry PsA risk alleles. Considering that RA and PsA depend on different molecular mechanisms, it is possible to hypothesize that the multiple drug resistance arises from the simultaneous dysregulation of several pathways implicated in both RA and PsA, even when the patient’s symptoms can be ascribed to a particular disease. Thus, a multitargeted approach, meaning the co-administration of drugs with different mechanisms of action, might help this selected cohort of individuals. In addition, the enrichment in the cohort of the rare HLA haplotype (B*14:02; C*08:02; DRB1*01:02; DQB1*05:01) and its previous association with other autoimmune diseases and hormone imbalance suggests that it may be involved in defining the patients’ phenotype (i.e., drug-resistant RA/PsA) [26]. 

Furthermore, the combination of SNPs-array and WES revealed the presence of likely pathogenic variants within 18 novel “drug-resistance” RA and PsA candidate genes in 10 patients (SP5 subject did not carry any suggestive variant). This is particularly relevant considering that few causative genes have been described for both RA and PsA. Indeed, despite the *PTPN22* gene being well known to be associated with the development of RA and PsA (especially in the European population) [1], none of the patients carry pathogenic variants of this gene, suggesting that other players might be involved. 

The most interesting novel “drug-resistance” RA and PsA candidate genes (*CYP21A2, UGT2B17*, *DVL1*, *PRKDC, ORAI1* and *MSR1)* are those in which a variant was identified in more than one patient. In detail, the *CYP21A2* gene is associated with the signalling of cortisol [27], while *UGT2B17* is involved in drug metabolism and sex hormone homeostasis [36]. To our knowledge, whether *UGT2B17* mediates DMARDs metabolism is unknown and cannot be excluded. Moreover, the role of cortisol and sex hormones in inflammation and in both RA and PsA has already been largely documented [54,55]. Indeed, even though RA and PsA are clinically different and show different pathogenesis and disease progression, they share some clinical and pathogenetic features being both autoimmune and inflammatory rheumatic disorders. In fact, some csDMARDs, as well as some b-/ts-DMARDs are effective in both diseases. Additionally, many genes involved in inflammation contain estrogen receptor binding elements, and previous studies indicated that DMARDs are more effective in men [54]. Furthermore, it has been reported that repository corticotropin injections (RCIs), which stimulate the release of cortisol, corticosterone and androgenic substances [53], may be beneficial for patients displaying refractory RA. Thus, it might be reasonable to administer RCI to patients carrying variants within these genes. 

Regarding *DVL1* and *PRKDC* genes, they regulate the Wnt and NF-kB pathways, which are essential for both RA and PsA but also for shaping anti-TNFα pharmacological effects [31,56,57]. These findings further highlight the importance of these pathways in RA and PsA and the suitability of administering therapies targeting them. Nevertheless, the patients did not benefit from anti-TNFα treatments, suggesting that the drugs could not restore the equilibrium of the targeted signalling pathways, likely due to the presence of the variants described above. 

Furthermore, homozygous variants within *PRKDC* and *ORAI1* genes have already been associated with immune dysfunction, such as severe forms of immunodeficiency and autoimmunity [58]. Thus, we hypothesize that the heterozygous variants detected in the patients could trigger immune dysregulation, and therefore, be involved in defining disease progression and, ultimately, the drug-resistant phenotype. These patients might benefit from therapies directed against these molecular targets considering that it has already been proven that MTX administration modulates *PRKDC* expression [32,59]. In particular, MTX causes an increase in *PRKDC* expression modulating NF-kB pathway activation, further supporting the gene involvement in RA and PsA pathogenesis and progression [57,59]. 

On the same note, *MSR1* has already been associated with autoimmunity and RA development [29,30]. Indeed, it has been proved that the gene regulates soluble autoantigen concentration in mouse models of autoantibody-dependent arthritis [30]. Thus, as suggested for *PRKDC* and *ORAI1*, additional studies may confirm *MSR1* as a novel molecular target for “difficult to treat” RA and eventually also for PsA.

Interestingly, the other genes in which private variants were detected are involved in the same pathways previously described for the common genes. Some examples refer to those candidates participating in the regulation of the Wnt and NF-kB pathways (i.e., *NCAPD3, NCAPD2, WNT10A, ABCB7, HACE1, SAMHD1* and *CCDC88C*) [37,38,40,46,48,52] or those associated with cortisol (*CYP27A1)* and sex hormone (*GNRHR* and *ESR1*) signalling. Among these last ones, the *ESR1* gene has already been associated with drug resistance, particularly to leflunomide [47]. In detail, it has been suggested that *ESR1* gene variants may affect estrogen receptor expression and thus have immunological consequences. Indeed, in-vitro studies with human macrophages proved that estrogens decrease leflunomide action by reducing the proapoptotic activity of the drug [47]. Thus, the SN5 patient’s variant might alter estrogen receptors expression, modulating not only leflunomide response but also having additional immunological consequences.

Moreover, as described in the literature for *PRKDC* and *ORAI1*, mutations within *SERPING1* can also cause severe forms of inflammatory and autoimmune disorders [42,60]. This gene has a role in the complement cascade, whose involvement in RA etiology has already been suggested [61,62]. Considering that, to date, drugs targeting the complement cascade are not administered to treat either RA or PsA, this could be a potential therapeutic option to be considered for selected patients. As an example, the molecular target of the drug garadacimab (phase three for other applications) is the *SERPING1* encoded protein [63].

On the same note, an additional variant was identified within *CLCF1,* which is involved in the signalling of crucial cytokines for both RA and PsA and targeted by already available therapies (i.e., tocilizumab and sarilumab) [10,11]. Interestingly, anti-IL6 drugs are employed to treat RA, but the *CLCF1* variant was identified in the PsA subject SN3. Thus, additional studies may confirm that a selected cohort of “difficult to treat” PsA patients may benefit from specific RA drugs (i.e., anti-IL6 drugs).

Additional studies involving other “difficult to treat” patients are needed to strengthen the association between the genes and phenotypes. Nevertheless, thanks to this multi-step approach, we hypothesize that the joint action of the peculiar HLA phenotypes and the several likely damaging variants detected in different genes may be responsible for the drug resistance. In conclusion, our strategy allowed for the discovery of 18 novel RA or PsA candidate genes and the definition of novel genotype-phenotype correlations to guide clinicians toward selecting the best therapeutic approach. 

## Figures and Tables

**Table 1 jpm-12-01618-t001:** Clinical features of the RA and PsA patients and the administered therapy. The table reports the main clinical features of each patient and the complete list of failed therapies. Failed b-DMARDs and ts-DMARDs are divided into three columns according to the reason that led to therapy discontinuation (e.g., primary inefficacy, secondary inefficacy, adverse effects requiring therapy suspension). (F = female; M = male; RA = rheumatoid arthritis; PsA = psoriatic arthritis; RF = rheumatoid factor; ACPA = anticitrullinated peptide/protein antibodies).

Subject (Gender)	Age (Age at Disease Onset)	Disease	Seropositivity	Autoimmune/ Inflammatory Comorbidities	Failed cs-DMARDs	b-/ts-DMARDs Primary Inefficacy	b-/ts-DMARDs Secondary Inefficacy	b-/ts-DMARDs Adverse Event
SP1 (F)	73 (60)	RA	RF + ACPA	Secondary Sjögren’s syndrome	methotrexate cyclosporine leflunomide	adalimumab etanercept baricitinib	abatacept	tocilizumab (neutropenia)
SP2 (F)	61 (47)	RA	RF + ACPA	-	methotrexate leflunomide	abatacept	rituximab certolizumab	etanercept (skin reaction)
SP3 (F)	60 (50)	RA	RF + ACPA	-	leflunomide methotrexate	rituximab tocilizumab	certolizumab adalimumab etanercept abatacept	baricitinib (serious infection)
SP4 (F)	54 (38)	RA	RF + ACPA	IgA nephropathy	methotrexate leflunomide sulphasalazine	adalimumab golimumab certolizumab abatacept baricitinib	etanercept tocilizumab	
SP5 (F)	48 (36)	RA	ACPA	-	methotrexate leflunomide	adalimumab abatacept baricitinib rituximab sarilumab	etanercept	tocilizumab (serious infection)
SN1 (F)	56 (36)	RA	Negative	Psoriasis	methotrexate leflunomide cyclosporine	etanercept rituximab abatacept tocilizumab	baricitinib adalimumab infliximab golimumab	anakinra (skin reaction)
SN2 (F)	60 (54)	RA	Negative	-	methotrexate leflunomide	tocilizumab	etanercept adalimumab	
SN3 (F)	35 (34)	PsA	Negative	-	cyclosporine	infliximab ixekizumab		adalimumab (paradoxical psoriasis)
SN4 (F)	29 (26)	PsA	Negative	-	methotrexate	infliximab adalimumab	secukinumab	
SN5 (M)	59 (48)	RA	Negative	-	methotrexate leflunomide	abatacept etanercept baricitinib anakinra tofacitinib	tocilizumab sarilumab adalimumab infliximab	
SN6 (F)	54 (27)	RA	Negative	-	methotrexate cyclosporine leflunomide sulphasalazine	etanercept tocilizumab abatacept	baricitinib infliximab adalimumab golimumab certolizumab	

**Table 2 jpm-12-01618-t002:** HLA typing for the genes HLA-B, HLA-C and HLA-DRB1. The table displays the HLA-B, HLA-C and HLA-DRB1 alleles carried by each patient. Alleles associated with RA development are reported in bold, while PsA/psoriasis risk alleles are underlined.

Subject	HLA-B	HLA-B	HLA-C	HLA-C	HLA-DRB1	HLA-DRB1
SP1	B*38:01P	B*41:01:01:01	C*12:03:01:01	C*17:01:01:05	**DRB1*10:01P**	DRB1*16:01:01
SP2	B*15:01:01:01	B*15:01:01:01	C*03:03P	C*03:04P	**DRB1*01:01P**	**DRB1*04:01P**
SP3	B*07:02P	B*18:01P	C*07:01P	C*07:02P	**DRB1*10:01P**	DRB1*14:01P
SP4	B*14:02:01:01	B*55:01P	C*03:03P	C*08:02P	**DRB1*01:02P**	**DRB1*10:01P**
SP5	B*08:01P	B*38:01P	C*07:01P	C*12:03:01:01	**DRB1*03:01P**	DRB1*08:03P
SN1	B*39:01P	B*45:01P	C*12:03:01:01	C*16:01:01:01	**DRB1*01:01P**	**DRB1*01:02P**
SN2	B*14:02:01:01	B*44:03P	C*04:01P	C*08:02P	**DRB1*01:02P**	DRB1*11:01P
SN3	B*14:02:01:01	B*57:01:01:01	C*06:02:01:01	C*08:02P	**DRB1*01:02P**	DRB1*07:01P
SN4	B*39:01P	B*41:02:01:01	C*12:03:01:01	C*17:03:01:01	DRB1*13:01P	**DRB1*13:03:01**
SN5	B*51:01P	B*56:01P	C*01:02P	C*15:02:01:01	**DRB1*04:08:01**	DRB1*11:04:01
SN6	B*27:02:01:01	B*35:01P	C*02:02:02:03	C*04:01:01:05	**DRB1*01:01P**	DRB1*16:01:01

**Table 3 jpm-12-01618-t003:** **Frequency of the patients’ HLA phenotypes in the world population**. The table displays the number of individuals with ten matches at five HLA loci for each patient. The definition of these numbers allowed us to calculate the frequency of the patients’ HLA haplotype in the world population.

Subject	N. of Donors with Ten Matches at 5 HLA Loci	Frequency (%)
SP1	1	2.50362 × 10^−6^
SP2	2	5.00723 × 10^−6^
SP3	5	1.25181 × 10^−5^
SP4	0	0
SP5	25	6.25904 × 10^−5^
SN1	1	2.50362 × 10^−6^
SN2	71	0.000177757
SN3	819	0.002050462
SN4	34	8.5123 × 10^−5^
SN5	0	0
SN6	1045	0.002616279

**Table 4 jpm-12-01618-t004:** Variants identified the RA/PsA cohort through WES. The table displays the variants detected in the patients. All variants were detected at the heterozygous state, except the one detected within the *UGT2B17* gene in patient SN4 (c.1158C>A, p.Tyr386*), which is indicated in the table with the asterisk. Genes reported in bold are those in which a variant was found in more than one patient. The *MSR1* gene is also reported in bold because, in addition to the frameshift variant carried by subject SP2, a large deletion affecting the gene was identified in patient SN3 through the SNPs-array technique. (AF = allele frequency; NA = not available; D = damaging; T = tolerated).

Subject	Gene	cDNA	ID	Protein	AF	PaPI Score	PolyPhen-2	DANN Score	SIFT
SP1	*WNT10A* (ENST00000258411)	c.321C>A	rs121908119	p.Cys107 *	0.00066	D	NA	D	NA
	***DVL1*** (ENST00000378888)	c.780C>G	NA	p.His260Gln	NA	D	T	D	D
SP2	*ABCB7* (ENST00000253577)	c.1767T>G	rs150273961	p.His589Gln	0.00056	D	D	D	D
	***MSR1*** (ENST00000350896)	c.260C>G	NA	p.Ser87 *	NA	D	NA	D	NA
SP3	***PRKDC*** (ENST00000314191)	c.7790G>A	rs55923149	p.Arg2597Gln	0.000026	D	D	D	D
	***UGT2B17*** (ENST00000317746)	c.1237G>T	NA	p.Ala413Ser	NA	D	NA	D	D
SP4	***DVL1*** (ENST00000378888)	c.1148C>A	NA	p.Ser383Tyr	NA	D	T	D	D
	***PRKDC*** (ENST00000314191)	c.5983G>A	rs768367219	p.Val1995Ile	0.000014	D	D	D	T
	***ORAI1*** (ENST00000330079)	c.144_150dupCGCCGTC	rs1555322554	p.Pro51fs	0.000014	D	NA	NA	NA
	*SERPING1* (ENST00000278407)	c.1198C>T	rs201363394	p.Arg400Cys	0.000049	D	D	D	T
SN1	***ORAI1*** (ENST00000330079)	c.698C>T	rs369586125	p.Pro233Leu	0.000012	D	T	D	D
SN2	*GNRHR* (ENST00000226413)	c.785G>A	rs104893837	p.Arg262Gln	0.0018	D	D	D	D
SN3	*NCAPD3* (ENST00000534548)	c.1712A>C	rs199812722	p.His571Pro	0.000069	D	D	D	T
	*CLCF1* (ENST00000312438)	c.226C>G	rs765776881	p.Pro76Ala	0.0000082	D	T	D	D
SN4	***UGT2B17*** (ENST00000317746)	c.1158C>A *	NA	p.Tyr386 *	NA	D	NA	D	NA
	*HACE1* (ENST00000262903)	c.217G>A	rs946488994	p.Ala73Thr	NA	D	T	D	D
SN5	*NCAPD2* (ENST00000315579)	c.860C>G	NA	p.Pro287Arg	NA	D	D	D	D
	*ESR1* (ENST00000440973)	c.715G>T	NA	p.Ala239Ser	NA	D	D	D	D
SN6	*SAMHD1* (ENST00000262878)	c.997C>T	rs770005027	p.Arg333Cys	0.000024	D	D	D	D
	*CYP27A1* (ENST00000258415)	c.1435C>T	rs72551322	p.Arg479Cys	0.0043	D	D	D	D
	*CCDC88C* (ENST00000389857)	c.5102G>T	NA	p.Arg1701Leu	NA	D	D	D	D

## Data Availability

The data presented in this study are available on request from the corresponding author. The data are not publicly available due to privacy restrictions.

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
