# Peer review of "High Throughput Genetic Characterisation of Caucasian Patients Affected by Multi-Drug Resistant Rheumatoid or Psoriatic Arthritis"

_jpm, 2022, doi:10.3390/jpm12101618_

Round 1
Reviewer 1 Report
This report is very elaborate and of high interest in the field of personalized medicine. However, it has important limitations. On the one hand, the sample of patients under study (difficult-to-treat) is very small (especially psoriatic arthritis) and a comparator group of RA and PsA that do respond to treatment has not been taken into account, in order to appreciate the differences genetics between the two groups, not only with control cohort composed of healthy individuals.
Reviewer 2 Report
1. The title of this paper was too long and may be corrected
2. In the part of introduction, failure of cDMARDs or biologics in PsA and RA should be enhanced. The duration of therapeutic failure should be made from previous papers.
3. The number of IRB should be given.
4. In the part of discussion, RA and PsA were different diseases with different pathogenesis or disease progression.
5. The finding of drug-resistance gene from RA and PsA was observed in this study, and the author should made adequate discussion about this finding.
Round 2
Reviewer 1 Report
The changes made are appropriate to the comments made.